Effects of rhodomyrtone on Gram-positive bacterial tubulin homologue FtsZ

Saeloh Dennapa 1 2
Wenzel Michaela 3
Rungrotmongkol Thanyada 4 5
Hamoen Leendert Willem 3
Tipmanee Varomyalin tvaromya@medicine.psu.ac.th 1 6
Voravuthikunchai Supayang Piyawan supayang.v@psu.ac.th 1 2
1 Excellence Research Laboratory on Natural Products, Faculty of Science and Natural Product Research Center of Excellence, Prince of Songkla University , Hat Yai , Thailand
2 Department of Microbiology, Faculty of Science, Prince of Songkla University , Hat Yai , Thailand
3 Bacterial Cell Biology, Swammerdam Institute for Life Sciences, University of Amsterdam , Amsterdam , Netherlands
4 Department of Biochemistry, Faculty of Science, Chulalongkorn University , Bangkok , Thailand
5 Center of Innovative Nanotechnology, Chulalongkorn University , Bongkok , Thailand
6 Department of Biomedical Science, Faculty of Medicine, Prince of Songkla University , Hat Yai , Thailand
Silva Pedro
Electronic publication date: 2017 Feb 2
Publication date: 2017
Volume: 5
Electronic Location ID: e2962
Received 2016 Nov 1; Accepted 2017 Jan 5
Copyright: ©2017 Saeloh et al.
Copyright year: 2017
Copyright holder: Saeloh et al.
License: This is an open access article distributed under the terms of the Creative Commons Attribution License, which permits unrestricted use, distribution, reproduction and adaptation in any medium and for any purpose provided that it is properly attributed. For attribution, the original author(s), title, publication source (PeerJ) and either DOI or URL of the article must be cited.
License URL: https://creativecommons.org/licenses/by/4.0/

Keywords: Rhodomyrtone, Tubulin homologue FtsZ, Cell division, Molecular dynamics simulation, Binding free energy

Funding: Higher Education Research Promotion and National Research University Project of Thailand, Office of Higher Education Commission TRF Senior Research Scholar RTA5880005 Overseas thesis research study from the Graduate School, Prince of Songkla University Netherlands Organization for Scientific Research STW-Vici 12128 This work was supported by Higher Education Research Promotion and National Research University Project of Thailand, Office of Higher Education Commission and TRF Senior Research Scholar grant (Grant No. RTA5880005) the Thailand Research Fund. DS was funded by a scholarship for an overseas thesis research study from the Graduate School, Prince of Songkla University. LWH was funded by the Netherlands Organization for Scientific Research (NWO, http://nwo.nl/en, STW-Vici 12128). The funders had no role in study design, data collection and analysis, decision to publish, or preparation of the manuscript.

==============================
Rhodomyrtone, a natural antimicrobial compound, displays potent activity against many Gram-positive pathogenic bacteria, comparable to last-defence antibiotics including vancomycin and daptomycin. Our previous studies pointed towards effects of rhodomyrtone on the bacterial membrane and cell wall. In addition, a recent molecular docking study suggested that the compound could competitively bind to the main bacterial cell division protein FtsZ. In this study, we applied a computational approach (in silico), in vitro, and in vivo experiments to investigate molecular interactions of rhodomyrtone with FtsZ. Using molecular simulation, FtsZ conformational changes were observed in both (S)- and (R)-rhodomyrtone binding states, compared with the three natural states of FtsZ (ligand-free, GDP-, and GTP-binding states). Calculations of free binding energy showed a higher affinity of FtsZ to (S)-rhodomyrtone (−35.92 ± 0.36 kcal mol−1) than the GDP substrate (−23.47 ± 0.25 kcal mol−1) while less affinity was observed in the case of (R)-rhodomyrtone (−18.11 ± 0.11 kcal mol−1). In vitro experiments further revealed that rhodomyrtone reduced FtsZ polymerization by 36% and inhibited GTPase activity by up to 45%. However, the compound had no effect on FtsZ localization in Bacillus subtilis at inhibitory concentrations and cells also did not elongate after treatment. Higher concentrations of rhodomyrtone did affect localization of FtsZ and also affected localization of its membrane anchor proteins FtsA and SepF, showing that the compound did not specifically inhibit FtsZ but rather impaired multiple divisome proteins. Furthermore, a number of cells adopted a bean-like shape suggesting that rhodomyrtone possibly possesses further targets involved in cell envelope synthesis and/or maintenance.

Introduction

Rhodomyrtone, extracted from Rhodomyrtus tomentosa leaves, displays potent activity against many Gram-positive bacteria including methicillin-resistant Staphylococcus aureus (MRSA) (Limsuwan et al., 2009). The activity is comparable to that of the last-defence antibiotics, vancomycin and daptomycin, and it is effective against recently emerging vancomycin-intermediate S. aureus strains (Leejae, Taylor & Voravuthikunchai, 2013). Thus, rhodomyrtone is an interesting new antibiotic candidate to challenge drug-resistant bacterial infections. Prior to further evaluation of its clinical potential, it is important to understand how it kills bacteria. Therefore, the molecular targets of rhodomyrtone need to be identified.

Despite several attempts to elucidate the mechanism of action of rhodomyrtone, its exact target has not yet been found. Proteomic and transcriptomic analyses of rhodomyrtone-treated MRSA pointed towards the cytoplasmic membrane and cell wall being affected (Sianglum et al., 2012; Sianglum et al., 2011; Visutthi, Srimanote & Voravuthikunchai, 2011). Similarly, abnormalities of both cell wall and cell membrane were revealed by transmission electron microscopy (Sianglum et al., 2011). However, a recent in silico screening for potential rhodomyrtone targets suggested that the compound could competitively bind to the main bacterial cell division protein FtsZ (Saeloh, Tipmanee & Voravuthikunchai, 2016). Consequently, earlier studies showed that rhodomyrtone caused MRSA cells to slightly enlarge, did not exhibit bacteriolytic activity, and did not promote leakage of proteins out of cells (Leejae, Taylor & Voravuthikunchai, 2013; Limsuwan et al., 2009). These occurrences were promoting the hypothesis that rhodomyrtone could indeed inhibit an intracellular target such as FtsZ (Adams et al., 2011; Kaul et al., 2013).

FtsZ, a homologue of eukaryotic tubulin, drives bacterial cell division by forming the constricting Z-ring (Pilhofer et al., 2011). To perform this task, the protein forms polymers, a process which is driven by its ability to hydrolyse GTP to GDP (GTPase activity). Following polymerization of FtsZ into the Z-ring, the membrane and cell wall constrict and form into a septum, which separates the two daughter cells (De Boer, Crossley & Rothfield, 1992). Consequently, FtsZ disassembles and GDP is released from FtsZ, which is then ready to bind a new GTP molecule and polymerize again (Natarajan & Senapati, 2013; Oliva, Trambaiolo & Löwe, 2007). This process is essential for bacterial cytokinesis and FtsZ is essential in all bacteria. Conditional mutants defective in cell division elongate into filaments. Despite these promising features, cell division has not yet been exploited as an antibiotic target in the clinic. Therefore, FtsZ is an interesting target for new anti-bacterial drugs.

Computer-aided techniques have become widespread in various areas of biological research. In particular, a combination of bioinformatics tools and experimental methods have been efficiently applied to uncover the mechanisms of effector molecules on their targets (Qiu et al., 2013; Singh et al., 2014). Currently, molecular dynamic (MD) simulation, a common computational technique used for studying proteins, is contributing to drug discovery and development. This approach is able to provide information at atomic levels by calculating the interactions between ligands and receptors and predicting conformational changes in drug-binding targets.

In order to analyse the inhibitory effects of rhodomyrtone on FtsZ in more detail, we used MD simulation to predict the interactions and impacts of the compound on the structure of FtsZ. Using polymerization (light scattering) and GTPase activity assays, we found that rhodomyrtone affected the function of purified FtsZ. Fluorescence light microscopy finally gave insights into the effect of rhodomyrtone on FtsZ in live bacteria.

Materials and Methods

Preparation of structures

In this study, we investigated the structure of FtsZ bound to various ligands as well as ligand-free FtsZ using molecular dynamics simulation. The preparation of the FtsZ starting coordinates was carried out using GDP-FtsZ from Staphylococcus aureus, (PDB entry 3VOA) as a molecular template. First, co-crystallized GDP in the structure was removed to obtain a ligand-free FtsZ starting structure. GDP, GTP, (S)-rhodomyrtone, and (R)-rhodomyrtone were docked to the ligand-free FtsZ protein in a molecular docking approach. GDP and GTP bound to FtsZ mimicked an in vivo state in S. aureus (Matsui et al., 2012) while both enantiomers of rhodomyrtone were used for assaying drug-target interaction with FtsZ. Since the structures of GTP and rhodomyrtone were artificial, ligand Cartesian coordinates were constructed and energy-minimized using ArgusLab 4.0.1 software (Thompson, 2004). Ligand-FtsZ complexes were created by molecular docking. Docking studies were carried out using the Autodock4 package (Morris et al., 2009) to predict the most convenient conformation and ligand position bound to the protein. A grid box of 110 Å × 110 Å × 110 Å with a grid spacing of 0.375 Å was established in the center of a macromolecule. A ligand was regarded to be a flexible molecule in search of the best position in the grid space of the rigid protein. Fifty independent docking jobs, each consisting of 200 runs, were conducted with a Lamarckian genetic algorithm employed with default parameters. A docked complex structure was chosen on the basis of the lowest binding energy. Finally, the five structures (GDP-FtsZ, GTP-FtsZ, (S)-rhodomyrtone-FtsZ, (R)-rhodomyrtone-FtsZ, and ligand-free FtsZ) were obtained to initiate molecular dynamic simulation.

Molecular dynamic simulation

Molecular dynamics of the above-mentioned FtsZ forms were conducted using the AMBER12 package (Gohlke, Kiel & Case, 2003) to observe ligand-induced conformational changes of FtsZ as well as the binding energy in dynamic conditions. Firstly, molecular information of both enantiomeric forms of rhodomyrtone, such as RESP atomic charges and bond parameters, was deducted from an optimized structure. Geometry optimizations and electrostatic charge calculations were performed using Gaussian 03 (Gaussian Inc., Wallingford, CT), and RESP charges were automatically generated using an antechamber program (Gohlke, Kiel & Case, 2003). Atomic charges of GDP and GTP were directly adopted from previous studies (Bayly et al., 1993). Regarding the FtsZ structure, the protonation states of all ionisable amino acid side chains were calculated at pH 7. In His10, a hydrogen at the side chain was located at the ε-nitrogen atom, and no doubly protonated histidine was applied in the FtsZ structure (Natarajan & Senapati, 2013). Missing hydrogen atoms were added by a leap program. All five protein complexes were solvated by a pre-equilibrated TIP3P water rectangular box with an edge of 12 Å. Potassium (K+) and chloride (Cl−) ions were added, yielding 150 mM of KCl solution. The system was energy-minimized for 2,000 steps using the steepest descent algorithm to remove improper van der Waals contacts, and continued with a 500 ps canonical (NVT) ensemble at a temperature of 310 K (37 °C) using a time step of 1 fs. Harmonic potential was applied in the NVT simulation for positional restraint of the protein and ligand, using force constants of 200, 100, 50, 25, and 10 kcal mol−1 Å−2 in each 100 ps, respectively. After 500 ps, the restraint on the protein and ligand components was released and the system was switched into an isobaric-isothermal (NPT) simulation at a constant pressure of 1 atm, and 310 K with a time step of 2 fs for 150 ns. In the NVT simulation, the temperature was controlled by a Langevin thermostat (Allen & Tildesley, 1987), while in the NPT simulation, the temperature and pressure were regulated using a weak-coupling algorithm (Berendsen et al., 1984). Finally, 1,000 snapshots from the last 50 ns of NPT trajectory were used to compute a configuration average and structural analysis.

Binding free energy analysis

To evaluate the relative binding affinity of rhodomyrtone to FtsZ, the Molecular Mechanics/Poisson Boltzmann Surface Area (MM/PBSA) approach (Kollman et al., 2000; Srinivasan et al., 1998) was chosen. In brief, the relative binding free energy (ΔG) can be computed from energetic differences as follows (Genheden & Ryde, 2015; Homeyer & Gohlke, 2012; Wang, Hou & Xu, 2006): ΔG=GLR−GL−GR.

GLR, GL, and GR represent the free energy components of the ligand–receptor complex, ligand, and receptor respectively. The free energy of each state was derived from molecular mechanics energy, broken down into: G=Ebond+Eel+EvdW+Enpl+Epol.

In the equation, the variables are as follows: Ebond, Eel, and EvdW are the MM energy values from the bonding terms (bond, angle, and dihedral), electrostatic, and van der Waals interactions, whereas Enpl and Epol are the nonpolar and polar contributions due to solvent solvation energy (Genheden & Ryde, 2015).

Since in this study all simulations were carried out using an MD trajectory, binding free energy was reported to be a configuration average 〈ΔG〉, obtained through the formula 〈ΔG〉= 〈GLR−GL−GR〉.

Herein, the binding free energy calculation was executed in a condition of 0.15 M salt concentration on 5,000 equidistant snapshots from a 100–150 ns MD trajectory. The calculation was performed using a Python script (MMPBSA.py) implemented in an AMBER12 package. The total binding free energy of the ligands (rhodomyrtone and GDP) as well as their energy contribution determined and were used to assess the binding affinity of the ligand to FtsZ.

Purification of rhodomyrtone

Rhodomyrtone was isolated from leaves of Rhodomyrtus tomentosa by extraction with 95% ethanol as described by our group (Hiranrat & Mahabusarakam, 2008; Limsuwan et al., 2009). Purity of rhodomyrtone was confirmed by nuclear magnetic resonance (NMR) and mass spectrometry (MS) (Hiranrat & Mahabusarakam, 2008; Salni et al., 2002). Purified rhodomyrtone was dissolved in dimethyl sulfoxide (DMSO, Merck, Germany) before use.

Purification of FtsZ

Calcium-competent Escherichia coli BL21 (DE3) cells were freshly transformed with the plasmids pCXZ and pBS58 for co-expression of untagged B. subtilis FtsZ with E. coli FtsQAZ, the latter helping E. coli to survive stress induced by overexpression of the B. subtilis protein (Wang & Lutkenhaus, 1993). Colonies were selected on Luria-Bertani (LB) agar plates supplemented with 50 µg/mL ampicillin (Sigma-Aldrich) and 50 µg/mL spectinomycin (Sigma-Aldrich). Overnight cultures inoculated with a single transformant colony were diluted 1:100 in fresh antibiotic-containing LB medium. 6 L of culture were grown at 37 °C to an OD600nm of 0.4 and induced with 1 mM IPTG (Sigma-Aldrich) for 4 h. Cultures were quickly cooled on slush ice, harvested by centrifugation, and washed once in 50 mM Tris pH 7.5, 100 mM NaCl, 1 mM EDTA (Sigma-Aldrich). The dry cell pellet was flash frozen in liquid nitrogen and stored at −80 °C until further use. The pellet was dissolved in 60 mL 50 mM Tris pH 8.0, 50 mM KCl, 1 mM EDTA, 10 mM MgCl2, 1 mg/mL DNase (Sigma-Aldrich), and 1 complete mini protease inhibitor tablet (Roche). The cells were disrupted by French Press and cell debris was removed by centrifugation (200,000 × g, 1 h). The supernatant was subjected to ammonium sulfate precipitation as follows. 26.4 mL of a saturated ammonium sulfate solution were added drop by drop under continuous stirring at 4 °C, followed by further stirring for 20 min. Precipitated proteins were removed by centrifugation (30,000 × g, 30 min) and the FtsZ-containing supernatant was subjected to a second precipitation step by adding 9.5 mL of saturated ammonium sulfate solution as described above. After 10 min of stirring, FtsZ was spun down (30,000 × g, 30 min) and the pellet was dissolved in 45 mL 50 mM MES-KOH pH 6.5, 5 mM MgCl2 (buffer A), followed by ion exchange chromatography. The sample was loaded onto a 5 mL HiTrap Q HP column (GE Healthcare) equilibrated with 3 column volumes of buffer A. The column was washed with buffer A until reaching a stable baseline, followed by washing with 5% buffer B (50 mM MES-KOH pH 6.5, 5 mM MgCl2, 1 M KCl). After reaching a stable baseline again, FtsZ was eluted in a gradient up to 50% buffer B over 5 column volumes. FtsZ-containing fractions were pooled and concentrated with 10 kDa molecular weight cutoff filters (Amicon), if necessary. Glycerol was added to a final concentration of 10% prior to flash freezing. Samples were stored as single-use aliquots until further use.

FtsZ polymerization in vitro

FtsZ polymerization was monitored by 90° light scattering. FtsZ (10 µM) was added in a Tris buffer (pH 7.4, 200 mM KCl, and 1 mM EDTA) in the absence (1% DMSO as a control) or presence of 10 µM rhodomyrtone, 20 µM rhodomyrtone, or 20 mM 3-methoxy-benzamide (3-MBA, Sigma-Aldrich), respectively. FtsZ assembly was started by addition of 5 mM MgCl2 and 1 mM GTP and monitored by observing light scattering over 2 min in a cuvette chamber at 37 °C. Measurements were performed with a PTI fluorometer operated under the control of FeliX32 software. Both the excitation and emission wavelengths were set at 350 nm and a slit width of 4 nm was used.

To visualise FtsZ filaments, 5 µl of sample were withdrawn immediately after starting the reaction and diluted 1:5 in the same buffer additionally containing 10% polyethylen glycol (Sigma-Aldrich) as a crowding agent. FtsZ filaments were visualised after 2 min of incubation at 30 °C using a Nikon Eclipse Ti microscope equipped with a CFI Plan Apochromat DM 100× oil objective, an Intensilight HG 130 W lamp, a C11440-22CU Hamamatsu ORCA camera, and NIS-Elements software, version 4.20.01.

Determination of GTPase activity

The GTPase activity of FtsZ was analyzed using a malachite green/ammonium molybdate assay as described by Bharat, Blanchard & Brown (2013) with a minor modification. FtsZ (10 µM) was incubated without or with different concentrations of rhodomyrtone (5 µM, 10 µM, and 20 µM) and 3-MBA (5 mM, 10 mM, and 20 mM) in 50 mM Tris buffer (pH 7.4) containing 200 mM KCl and 1 mM EDTA at 37 °C for 10 min. The hydrolysis reaction was initiated by addition of 5 mM MgCl2 and 1 mM GTP. After 5 min of incubation, HClO4 (10% v/v) was added to quench the reaction. 100 µl of each sample was transferred to a 96-well plate, mixed and incubated with 40 µl of reaction agent for 2 min. Released inorganic phosphates were monitored by measuring the absorbance at 600 nm. The amount of released phosphate was calculated using a phosphate standard curve prepared with KH2PO4.

Fluorescence microscopy

To investigate the effect of the compound on Z-ring formation, FtsA, and MinD localization in living cells, B. subtilis 874 (Weart et al., 2005) expressing green fluorescent protein (GFP)-tagged FtsZ, PG62 (Gamba et al., 2009) expressing FtsA fused to yellow fluorescent protein (YFP), and 4181 (Hamoen et al., 2006) expressing gfp-sepF were grown overnight in LB broth at 30 °C in the presence of 50 µg/mL spectinomycin (Sigma-Aldrich). The overnight culture was diluted 1:100 into LB containing 0.5% xylose (FtsZ), 0.15% xylose (SepF), or 0.1 mM isopropyl β-D-1-thiogalactopyranoside (IPTG) (FtsA) to induce expression of GFP or YFP fusion proteins, and incubated at 30 °C until an OD600nm of 0.3. The cultures were then treated with 1xMIC (0.5 µg/mL, 1.13 µM), 2xMIC (1 µg/mL, 2.26 µM), and 4xMIC (2 µg/mL, 4.52 µM) of rhodomyrtone, 4xMIC (2 mg/mL, 13.23 µM) of 3-MBA, 100 µM CCCP (Sigma Aldrich), and 1% DMSO as a negative control. At various time pionts after treatment, samples were placed onto 1% agarose slides and images were taken using a Nikon Eclipse Ti microscope as specified above.

Morphological changes observed with phase-contrast microscopy

B. subtilis 168 (Anagnostopoulos & Spizizen, 1961) overnight cultures were diluted 1:100 in fresh LB and aerobically grown at 37 °C until an OD600nm of 0.3. Then the cultures were split-treated with 1xMIC (0.5 µg/mL, 1.13 µM), 2xMIC (1 µg/mL, 2.26 µM), and 4xMIC (2 µg/mL, 4.52 µM) of rhodomyrtone, 1xMIC (0.5 mg/mL, 1.13 µM), 2xMIC (1 mg/mL, 2.26 µM), and 4xMIC (2 mg/mL, 13.23 mM) of 3-MBA, and 1% DMSO as a negative control. After 1 h, 2 h, and 4 h of incubation, samples were placed onto 1% agarose slides and imaged with an Olympus BX60 microscope equipped with a Photometrics CoolSNAP fx digital camera. Images were analyzed with Image J.

Results

Conformational changes of apo, GDP-, GTP-FtsZ in complex with (S)-rhodomyrtone and (R)-rhodomyrtone

To get insights into the molecular interactions of rhodomyrtone with FtsZ, we compared the structures of FtsZ bound to rhodomyrtone, in both its (R)- and (S)-enantiomeric forms (Fig. 1), with the natural states of FtsZ, namely the nucleotide-free and two nucleotide-bound forms (GDP and GTP). An FtsZ crystal structure from S. aureus (Matsui et al., 2012) (PDB code 3VOA with a resolution of 1.73 Å) was used as the representative of FtsZ. Molecular docking was employed to identify the most favorable position of the respective ligands, namely GDP, GTP, (S)-rhodomyrtone, and (R)-rhodomyrtone, upon interactions with FtsZ. A ligand-free protein was set as a ligand-free-form (apo-form). MD simulations were assessed by Amber12 Force Field for 150 ns, and root mean square distances (rmsd) of backbone atoms compared to the starting conformations were computed to analyze structural changes. Figure 2 illustrates that all systems approached an equilibrium after 50 ns and remained constant from around 100 ns. Thus, we selected the last 50 runs to determine the average rmsds of FtsZ in the apo-, GDP-, and GTP-bound forms, which were found to be 1.89, 1.85, and 2.20 Å, respectively. The values of all three states did not differ significantly, implying structural similarity among FtsZ forms in all states. Small variations are known to occur between monomer structures of apo-FtsZ, GDP-FtsZ, and GTP-FtsZ across various bacterial species and nucleotide-binding states (Hsin, Gopinathan & Huang, 2012). (R)-rhodomyrtone-FtsZ and (S)-rhodomyrtone-FtsZ were found with rmsd values of 1.68 and 2.58 Å, respectively.

Figure 1 Rhodomyrtone structure.

(A) (S)-rhodomyrtone and (B) (R)-rhodomyrtone.

Figure 2 Cα root-mean-square distance (rmsd) values of simulated FtsZ structures as a function of time.

Rmsds were calculated by superposing each snapshot on the starting structure to remove the rigid body translations and rotations. (A) Rmsds for total simulation time of 150 ns. (B) Rmsds for last 50 ns of simulation.

Figure 3 Average structures of each FtsZ state.

(A) The reference structure of FtsZ from PDB: 3VOA (Matsui et al., 2012), (B) Ligand-free state of FtsZ: blue, (C) GDP-FtsZ complex: orange, (D) GTP-FtsZ complex: magenta, (E) (S)-rhodomyrtone-FtsZ complex: red, (F) (R)-rhodomyrtone-FtsZ complex: pink.

To elucidate conformational changes in each region of the simulated proteins, time-averaged structures were created and compared with the starting crystal conformation (Fig. 3A). None of the averaged structures displayed significant conformational changes. Ligand-free FtsZ (Fig. 3B) appeared similar in overview; however, the loop T3 and loop T7 regions were slightly different from the original structure. The structural changes in the GDP- and GTP-binding states (Figs. 3C–3D) showed changes in loop H6 and loop H7, with this loop bent towards the substrate binding pocket. Loop H6 and loop H7 are important for binding one FtsZ monomer to another to finally form FtsZ filaments (Matsui et al., 2014). Therefore, conformational changes in this region may reflect FtsZ polymerization. The average structures of the (S)-rhodomyrtone and (R)-rhodomyrtone binding states exhibited differences in terms of their binding positions. The (S)-rhodomyrtone binding state (Fig. 3E), in which the compound was located in the same region as the natural substrate, showed an alteration in the substrate binding pocket at loop T4. While the loop moved into the pocket, the helix H7 tilted away from the original structure, resulting in opening of the binding pocket. Additionally, in the (S)-rhodomyrtone state, we observed differences from the GDP- and GTP-binding states in loop H6 and loop H7. The changes in this pocket area might influence the S7-S10 region, harboring the hydrolase domain. Surprisingly, although the (R)-rhodomyrtone binding site is located in the GTPase domain, no obvious structural alterations were observed (Fig. 3F).

Calculation of free binding energy

To estimate effective binding energy, energetic analysis was performed using the Molecular Mechanics/Poisson Boltzmann Surface Area (MM/PBSA) method as summarized in Table 1. Electrostatic forces significantly contributed to the binding of GDP to FtsZ as evidenced by the strong interaction between the beta-phosphate of GDP and arginine (Arg135) at the binding site. Nonpolar interactions played a crucial role in binding (S)-rhodomyrtone to the FtsZ protein. In the simulated model (Fig. 4A), (S)-rhodomyrtone interacts with a phenylalanine residue in a hydrophobic pocket of FtsZ, which could be observed between the N-base aromatic ring of GDP and aromatic residues of FtsZ (Huecas et al., 2015). This implies that an aromatic ring of (S)-rhodomyrtone could interact with phenylalanine via π–π stacking (Fig. 4B). The total free binding energies of the GDP, (S)-rhodomyrtone, and (R)-rhodomyrtone-FtsZ complexes were −23.47 ± 0.25, −35.92 ± 0.36, and −18.11 ± 0.11 kcal mol−1, respectively. These values reflect a higher affinity of FtsZ to (S)-rhodomyrtone than the GDP substrate. Interestingly, less affinity was observed in the case of (R)-rhodomyrtone.

Table 1 Free energy components and total binding free energies (kcal mol−1).

Compound	GDP	(S)-rhodomyrtone	(R)-rhodomyrtone	
ΔEvan der Waal	−40.41 ± 0.20	−45.48 ± 0.12	−29.04 ± 3.30	
ΔEelectrostatic	−171.55 ± 1.26	0.47 ± 0.15	−18.40 ± 5.83	
ΔGpolar	192.85 ± 1.18	20.69 ± 0.15	32.70 ± 5.70	
ΔGnon-polar	−4.51 ± 0.01	−4.26 ± 0.01	−3.37 ± 0.25	
ΔGbinding	−23.61 ± 0.33	−35.92 ± 0.36	−18.11 ± 3.67	

Figure 4 Structural model of rhodomyrtone-bound FtsZ.

(A) FtsZ crystal structure from S. aureus, PDB 3VOA, comprising two subdomains: (i) C-terminal (nucleotide binding region) colored yellow, (ii) N-terminal (GTPase region) colored blue. (S)-rhodomyrtone is shown in red and (R)-rhodomyrtone in purple. (B) Residues of the protein within 3 Å of (S)-rhodomyrtone. The structure of (S)-rhodomyrtone is rendered in balls and sticks and the atoms are colored according to their types: C-ice blue, O-red.

Molecular dynamics of the ligand-FtsZ states

To observe motion in each FtsZ region, root-mean-square fluctuations (rmsf) were computed from trajectories of the last 50 ns simulations. Rmsf (Fig. 5) indicates the flexibility of each amino acid residue in the protein (307 amino acids). Several regions of ligand-free FtsZ, such as T3, T5, H6-10, and S9-10, have been reported to be more flexible than GDP or GTP-bound states (Natarajan & Senapati, 2013). The FtsZ structure can generally be divided into two major parts separated by H6-7 helices: (i) a nucleotide binding region, and (ii) a region responsible for hydrolyzing the nucleotide bound to the neighboring FtsZ molecule (Hurley et al., 2016). The rmsf values in the GDP- and GTP-binding states differed in the nucleotide binding area and the H1, T4, and H6-8 regions. Smaller rmsf values in the GTP-binding state imply less flexibility of the protein structure. The reduced plasticity of the GTP-binding states may be due to binding of its gamma-phosphate to FtsZ, possibly leading to a straight filament formation in contrast to the normally curved filaments induced by hydrolysis of GTP to GDP (Hsin, Gopinathan & Huang, 2012). The state of (S)-rhodomyrtone, bound to FtsZ in its nucleotide binding region, exhibited rmsf values close to the values of the GDP- and GTP-binding states at loop T3, which is the binding site of the nucleotide. Interestingly, loops T4 and T5 of the (S)-rhodomyrtone binding state were more flexible than those of both nucleotide binding states. The results indicate that (S)-rhodomyrtone might affect FtsZ assembly dynamics. In contrast, the flexibility pattern in the (R)-rhodomyrtone-bound state was similar to that of ligand-free FtsZ (Fig. S1). This finding further supports the interaction analysis, presented in Table 1, suggesting that the (R)-enantiomer of rhodomyrtone displays weaker binding to FtsZ. The dynamic behaviour of (R)-rhodomyrtone-bound FtsZ was not different from the ligand-free protein. Our findings suggest an enantiomeric specificity of rhodomyrtone towards the FtsZ protein.

Figure 5 Fluctuations of FtsZ in four states of FtsZ depicted by root-mean-square-fluctuations (rmsf) of the Cα atoms in three states of FtsZ monomer.

In vitro effects of rhodomyrtone on FtsZ

We further investigated the effects of rhodomyrtone on FtsZ in vitro and in vivo. To this end, we used B. subtilis, a Gram-positive model organism for the study of cell division (Pinho, Kjos & Veening, 2013). B. subtilis FtsZ shares 70% amino acid sequence homology with S. aureus protein, in line with studies on other compounds (Mathew et al., 2011; Sass et al., 2011). Rhodomyrtone was highly active against both organisms. We performed 90° angle light scattering and phase-contrast microscopy using purified B. subtilis FtsZ. Rhodomyrtone, a natural compound containing (S)-rhodomyrtone and (R)-rhodomyrtone, reduced FtsZ assembly in a concentration-dependent manner by maximally 36%. Meanwhile, 3-MBA, a stabilizer of FtsZ polymers (Adams et al., 2011), enhanced polymerization by 16% (Fig. 6A). This is in line with our in silico data suggesting competitive binding to the nucleotide binding pocket, which should reduce the formation of filaments.

Figure 6 Effect of rhodomyrtone on FtsZ assembly in vitro.

(A) GTP-induced polymerization of purified FtsZ exposed to 1% DMSO (a negative control), 10 µM rhodomyrtone, 20 µM rhodomyrtone, and 20 mM 3-MBA. The experiment was performed in triplicate and average spectra are shown. Error bars show standard errors of the mean. (B) Phase-contrast images of FtsZ bundles from the same samples.

To confirm that rhodomyrtone only affected FtsZ bundling, instead of causing any aberrant bundle formation or nonspecific aggregation of the protein, we examined the same samples microscopically. We neither observed protein aggregation, nor any obvious differences in the appearance of FtsZ bundles when exposed to rhodomyrtone or 3-MBA, compared with FtsZ in 1% DMSO (Fig. 6B). It has to be noted that under our experimental conditions, 20 mM of 3-MBA were necessary to observe any effect in light scattering experiments. In contrast to some of its optimized derivatives, 3-MBA has been reported to be only a weak inhibitor of FtsZ (Czaplewski et al., 2009; Ohashi et al., 1999) and rhodomyrtone in fact displays a better binding energy and inhibitory constant to FtsZ than 3-MBA (Saeloh, Tipmanee & Voravuthikunchai, 2016).

FtsZ assembly is dependent on GTPase activity (Mukherjee & Lutkenhaus, 1998) and our molecular docking studies suggested competitive binding of (R)-rhodomyrtone to the GTPase domain. In addition, conformational changes in the GTPase domain were observed in (S)-rhodomyrtone binding FtsZ. Therefore, we determined the effect of rhodomyrtone on GTPase activity using malachite green. GTPase activity was decreased up to 45% in the presence of rhodomyrtone, similar to the inhibition achieved with 3-MBA (Fig. 7). In contrast, cephalexin, an inhibitor of penicillin-binding proteins active at the cell division site, served as a negative control and did not affect GTPase activity.

Figure 7 Effect of rhodomyrtone on the GTPase activity of purified FtsZ.

GTP hydrolysis was carried out in the presence of rhodomyrtone (5, 10, and 20 µM) or 3-MBA (5, 10, and 20 mM) by adding 1 mM GTP. Cephalexin (10 µM) was used as a negative control. The experiment was performed in triplicate. Error bars show standard errors of the mean.

FtsZ polymers disassemble when nucleotide molecules cannot reach FtsZ (Arjes et al., 2015). Due to better affinity of rhodomyrtone, it could be speculated that (S)-rhodomyrtone, binding to the nucleotide binding site of FtsZ, can compete with GDP in a monomeric FtsZ, thereby obstructing replacement of GTP and consequently reducing FtsZ assembly. In addtion, it was found that the GTPase region was changed by (S)-rhodomyrtone, supporting the influence on GTPase activity of FtsZ. Recently, several approaches have been undertaken to develop FtsZ inhibitors, and most of the molecules targeting the substrate binding region have shown similar effects on FtsZ assembly and GTPase activity (Schaffner-Barbero et al., 2011). However, we have to consider that rhodomyrtone is a racemic mixture composed of (S)-rhodomyrtone and (R)-rhodomyrtone. Therefore, this may cause partial effect on FtsZ.

In vivo effects on Z-ring formation

We further investigated whether rhodomyrtone could affect FtsZ in B. subtilis in vivo. FtsZ and its membrane anchor FtsA are the first proteins located at the middle of the cell and recruit other proteins involved in cell division (Adams & Errington, 2009). Therefore, mid-cell localization of FtsZ is crucial for the cell division process. We microscopically examined the presence of Z-rings at mid-cell using a GFP fusion to FtsZ. In the absence of antibiotics (Figs. 8A and 8B), FtsZ clearly localized at mid-cell. Treatment with 3-MBA (Figs. 8C and 8D) resulted in rapid perturbation of FtsZ localization into a number of foci. In contrast, treatment with rhodomyrtone resulted in a diffuse cytosolic GFP signal. Importantly, this effect of rhodomyrtone on FtsZ localization was not observed at inhibitory concentrations (Figs. 8E and 8F), but only after 1 h of treatment with 2xMIC (Figs. 8G and 8H) or 30 min of treatment with 4xMIC (Figs. 8I and 8J), a concentration already causing cell lysis of B. subtilis. To examine whether rhodomyrtone specifically affects the localization of FtsZ or other divisome proteins as well, we next tested its effect on the localization of FtsA and SepF, two peripheral membrane proteins that anchor the Z-ring to the membrane. Similarly to what was observed for FtsZ, FtsA was detached from the membrane after either 60 min of treatment with 2xMIC or 30 min of treatment with 4xMIC of rhodomyrtone (Figs. 9E–9H ). SepF was already delocalized after only 10 min of treatment with 2xMIC of rhodomyrtone (Fig. 10). 3-MBA had no effect on localization of either FtsA or SepF (Figs. 9I–9J and Fig. 10). Thus, in contrast to 3-MBA rhodomyrtone does not specifically affect FtsZ but multiple divisome proteins.

Figure 8 Effect of rhodomyrtone on FtsZ localization.

B. subtilis 874 expressing GFP-FtsZ was exposed to (A) 1% DMSO for 30 min, (B) 1% DMSO for 60 min, (C) 4xMIC of 3-MBA for 30 min, (D) 4xMIC of 3-MBA for 60 min. (E) 1xMIC of rhodomyrtone for 30 min, (F) 1xMIC of rhodomyrtone for 60 min, (G) 2xMIC of rhodomyrtone for 30 min, (H) 2xMIC of rhodomyrtone for 60 min, (I) 4xMIC of rhodomyrtone for 30 min, and (J) 4xMIC of rhodomyrtone for 60 min.

Figure 9 Localization of FtsZ and FtsA in comparison with CCCP.

B. subtilis874 expressing GFP-FtsZ was exposed to (A) 1% DMSO for 10 min or (B) 100 µM CCCP for 10 min. B. subtilis PG62 expressing YFP-FtsA was exposed to (C) 1% DMSO for 10 min, (D) 100 µM CCCP for 10 min, (E) 2xMIC of rhodomyrtone for 30 min, (F) 2xMIC of rhodomyrtone for 60 min, (G) 4xMIC of rhodomyrtone for 30 min, (H) 4xMIC of rhodomyrtone for 60 min, (I) 4xMIC of 3-MBA for 30 min, and (J) 4xMIC of 3-MBA for 60 min.

Figure 10 Localization of SepF.

B. subtilis 4181 expressing GFP-SepF was exposed to 1% DMSO (negative control), 100 µM CCCP, 2xMIC of rhodomyrtone, 4xMIC of rhodomyrtone, or 4xMIC of 3-MBA for 10 or 30 min, respectively.

Dissociation of FtsZ into the cytosol is regularly observed with compounds affecting the membrane potential (Müller et al., 2016; Strahl & Hamoen, 2010; Te Winkel et al., 2016). This is because its membrane anchors FtsA and SepF require the membrane potential for membrane binding (Strahl & Hamoen, 2010). In line, treatment with the depolarizing ionophore CCCP resulted in delocalization of FtsZ (Fig. 9B), FtsA (Figs. 9C and 9D), and SepF (Fig. 10). Although delocalization of FtsZ and FtsA happened faster with CCCP, the ionophore caused the same phenotype as high concentrations of rhodomyrtone. Furthermore, both FtsA and SepF are considered reporter proteins for membrane depolarization (Strahl & Hamoen, 2010), suggesting that membrane dissociation of FtsZ caused by rhodomyrtone is due to delocalization of its membrane anchors, which is mediated by impairment of the cytoplasmic membrane.

Morphological changes

If inhibition of cell division through delocalization of multiple divisome proteins is in fact the principal mechanism of the compound, long-time treatment should result in cell elongation (Hwang & Lim, 2015; Schaffner-Barbero et al., 2011). Therefore, we examined the impact of rhodomyrtone on B. subtilis morphology by phase contrast microscopy. 3-MBA-treated cells were clearly longer than control cells while no cell elongation was observed with different concentrations of rhodomyrtone (Table 2). Instead, many cells appeared shorter and thicker. Interestingly, a number of cells adopted a bean-like shape (Fig. 11), suggesting that rhodomyrtone does not specifically inhibit cell division but might have additional cell envelope targets.

Table 2 The average cell length of B. subtilis cells in various times and treatments.

Compounds	Cell length (µm) a	
	1 h	2 h	4 h	
1% DMSO	8.84 ± 3.46	9.12 ± 5.46	5.08 ± 1.67	
1xMIC rhodomyrtone	8.88 ± 5.10	6.03 ± 2.18	4.56 ± 1.35	
2xMIC rhodomyrtone	7.34 ± 2.79	5.55 ± 3.61	5.23 ± 2.70	
4xMIC rhodomyrtone	6.67 ± 3.15	5.78 ± 2.89	4.44 ± 2.17	
4xMIC 3-MBA	13.76 ± 6.60	30.30 ± 12.00	NA	
Notes.

NA not applicable.

a ImageJ was used to measure the cell length of 100 cells of each condition.

Figure 11 Cell morphology of Bacillus subtilis 168.

Cells were incubated with 1% DMSO (negative control), different concentrations of rhodomyrtone (1xMIC, 2xMIC, and 4xMIC), and 4xMIC of 3-MBA. Pictures were taken after 1 h, 2 h, and 4 h. Phase-contrast images were obtained using an Olympus BX 50 microscope.

Discussion

In this study, we investigated the potential of rhodomyrtone to interact with and inhibit the essential cell division protein FtsZ. FtsZ, which has been found regulated in both transcriptome and proteome analyses of rhodomyrtone-treated S. aureus (Sianglum et al., 2012; Sianglum et al., 2011), has recently been identified as a possible molecular target in a preceding in silico study. This was corroborated by observation that S. aureus cells were slightly enlarged after prolonged treatment with rhodomyrtone (Saeloh, Tipmanee & Voravuthikunchai, 2016). Using molecular modeling, we could show that rohodomyrtone most likely binds to the nucleotide binding pocket of FtsZ, whereby (S)-enantiomer was more effective than (R)-enantiomer. Competitive binding of (S)-rhodomyrtone to the GDP/GTP-binding site should result in inhibition of FtsZ polymerization. In fact, in vitro experiments with purified FtsZ revealed that both polymerization and GTPase activity were affected by rhodomyrtone. However, polymerization was only inhibited by 36% while GTPase activity was reduced by maximally 45%. It is possible that rhodomyrtone as we used it, i.e., isolated from the natural source containing both (S)- and (R)-enantiomers, is not optimally efficient in inhibiting FtsZ. A similar phenomenon was observed with citronellal (Altshuler et al., 2013), a major component of Corymbia citriodora and Cymbopogon nardus essential oils. (+)-citronellal caused disruption of animal and plant microtubules, while (−)-citronellal did not. Similarly, the (R)-enantiomer of N-benzyl-3-sulfonamidopyrrolidine was effective in inhibiting polymerization of E. coli FtsZ, whereas (S)-enantiomer did not have any effect (Mukherjee et al., 2007). Moreover, the potency of small molecule FtsZ inhibitors was shown to be improved by eliminating enantiomeric conversion (Stokes et al., 2013). Therefore, future studies could rule out whether pure (S)-rhodomyrtone might be a more efficient inhibitor of FtsZ.

Despite being able to inhibit FtsZ in vitro, rhodomyrtone did not specifically inhibit FtsZ in B. subtilis in vivo. Localization of FtsZ was not affected by inhibitory rhodomyrtone concentrations, showing that FtsZ is not the main in vivo target. The protein was dispersed in the cytosol after treatment with higher concentrations of rhodomyrtone. Similar effects are typically observed with antibiotics that dissipate the membrane potential (Araújo-Bazán et al., 2016; Müller et al., 2016; Strahl & Hamoen, 2010; Te Winkel et al., 2016). This is because FtsZ is anchored to the membrane by two peripheral membrane proteins, FtsA and SepF. Both proteins require the membrane potential for membrane binding and delocalize into the cytosol upon its dissipation, which in turn results in cytosolic dispersion of FtsZ (Strahl & Hamoen, 2010). In fact, we could show that rhodomyrtone, similarly to CCCP and in sharp contrast to 3-MBA, does affect the localization of both FtsA and SepF as well, suggesting that the compound has a more general effect on the divisome than specifically inhibiting FtsZ polymerization. Considering that both FtsA and SepF are reporters for membrane depolarization (Strahl & Hamoen, 2010), it is likely that this is due to effects of rhodomyrtone on the cell membrane. However, it took longer treatment times and supra-inhibitory concentrations of rhodomyrtone to achieve delocalization of cell division proteins, showing that its mechanism must be different from ionophores like CCCP. Furthermore, long-time treatment of B. subtilis with rhodomyrtone did not result in cell elongation, which would be expected from an FtsZ inhibitor (Bhattacharya et al., 2013; Duggirala et al., 2014). A recent microscopy study determined the cytological profile of FtsZ inhibitors and identified at least three-fold cell elongation and either increased or decreased FtsZ ring spacing as essential factors to identify an FtsZ inhibitor (Araújo-Bazán et al., 2016). Neither of these was observed with rhodomyrtone. Instead, the compound led to bean-like cell deformations, indicating that rhodomyrtone has other targets involved in cell envelope synthesis or maintenance and does not specifically inhibit Z-ring formation. This is well in line with our observation that membrane potential-sensitive peripheral membrane proteins are affected by the compound, suggesting that it rather impairs cytoplasmic membrane function. In fact, fractionation experiments with S. aureus showed that rhodomyrtone is unable to cross the cytoplasmic membrane barrier to reach the cytosol in the first 4 h of treatment but instead accumulates in the cell debris (Data S3). Thus, it is reasonable to assume that the compound, although being able to inhibit FtsZ in vitro, does not reach this target in the in vivo situation. Instead, it is likely that rhodomyrtone interacts with membrane or cell wall-bound protein targets or with membrane or cell wall structural components, and delocalization of divisome proteins is a result of this interaction. Thus, cell shape deformations could be due to inhibition or de-regulation of cell wall synthesis enzymes such as penicillin-binding proteins or proteins involved in regulation of this process such as autolysins. An earlier transcriptomic study found upregulation of a number of genes encoding membrane or lipoproteins after treatment of S. aureus with rhodomyrtone (Sianglum et al., 2012) suggesting that it might interfere with the cytoplasmic membrane. In fact, it has recently been shown that daptomycin, which also leads to cell shape deformations, affects the cell wall synthesis machinery by disturbing membrane organization (Müller et al., 2016). However, the same transcriptome study as well as earlier proteomic studies on rhodomyrtone-treated S. aureus (Sianglum et al., 2011; Visutthi, Srimanote & Voravuthikunchai, 2011) showed the absence of typical cell envelope stress responses as elicited by e.g. daptomycin (Muthaiyan et al., 2008; Poole, 2012), indicating that the mode of action of rhodomyrtone is profoundly different from other compounds. Elucidation of rhodomyrtone’s effects on the bacterial cell envelope will be investigated in future studies.

Conclusions

We show that rhodomyrtone does inhibit FtsZ in vitro, probably by interaction of (S)-enantiomer with the nucleotide biding pocket. However, FtsZ is not the main in vivo target of the compound, which interferes with the localization of several divisome proteins and also seems to have an impact on the synthesis and/or maintenance of the cell envelope.

Supplemental Information

Figure S1 Fluctuations of FtsZ in four states of FtsZ depicted by root-mean-square-fluctuations (rmsf) of the Cα atoms in three states of the FtsZ monomer

Black for ligand-free FtsZ, blue for GTP-FtsZ, green for GDP-FtsZ, red for (S)-rhodomyrtone-FtsZ, and yellow for (R)-rhodomyrtone. The regions with the most significant changes are labeled.

Click here for additional data file.

Data S1 Phosphate standard curve prepared with KH2PO4 and raw data applied for GTPase activity analysis and preparation for Fig. 7

Click here for additional data file.

Data S2 Parameter files, library files, and the last snapshot of all FtsZ states

Click here for additional data file.

Data S3 Localization of rhodomyrtone in S. aureus ATCC 29213 after treatment for 4 h

Click here for additional data file.

The authors express gratitude to the Eclipse Computer Cluster, National Center for Genetic Engineering and Biotechnology (BIOTEC), Thailand. VT would like to thank Mr David Patterson of the International Affairs Office, Faculty of Medicine, Prince of Songkla University for manuscript proofreading and language editing service.

Additional Information and Declarations

Competing Interests

Author Contributions

Data Availability

The authors declare there are no competing interests.

Dennapa Saeloh and Michaela Wenzel conceived and designed the experiments, performed the experiments, analyzed the data, wrote the paper, prepared figures and/or tables.

Thanyada Rungrotmongkol performed the experiments.

Leendert Willem Hamoen contributed reagents/materials/analysis tools, reviewed drafts of the paper.

Varomyalin Tipmanee conceived and designed the experiments, performed the experiments, analyzed the data, contributed reagents/materials/analysis tools, wrote the paper, prepared figures and/or tables, reviewed drafts of the paper.

Supayang Piyawan Voravuthikunchai conceived and designed the experiments, analyzed the data, contributed reagents/materials/analysis tools, reviewed drafts of the paper.

The following information was supplied regarding data availability:

The raw data has been supplied as a Supplemental File.

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
