# Peer review of "Effects of rhodomyrtone on Gram-positive bacterial tubulin homologue FtsZ"

_PeerJ, doi:10.7717/peerj.2962_

## Round 0.1 · original submission · Major Revisions

· Academic Editor

Major Revisions

Please address all issues highlighted be the reviewers.

Review-level observations by the editor:

- I would like to know how the MM/PBSA affinity you computed for rhodomyrtone compares to GTP affinity computed by the same methods. That should not be much work, as you have already run a GTP-bound MD simulation. As they stand, your computational results imply that rhodomyrtone-FstZ complexes dissociate less that GDP-FstZ but do not show that rhodomyrtone binds more strongly that GTP. Computation of the GTP binding affinity might help explain the apparent lack of rhodomyrtone effect in vivo (if you find that affinity for rhodomyrtone is not much higher than the affinity for GTP)

- Please explain the rationale for the e-protonation of His10

Reviewer 1 ·

Basic reporting

No comments except my comment #2 to authors below. I do not know if Molecular Dynamics trajectories count as raw data that must be made available.

Experimental design

No comment

Validity of the findings

No comments, except for #4 and 5 below - authors report calculated binding affinities and then experimental results that do not support the calculated binding affinity. They do not explain whether the calculated binding affinities should translate to experimental results, although the implication that they will is strong. They also do not provide context for understanding what these numbers mean if they are not expected to be reflected in experiment. They also should make some attempt to comment on the disagreement between experiment and calculation, other than just saying that they disagree. I do not think that these are hard questions to ask, and they should not require further experiments or calculations; merely explanations.

Comments for the author

This paper reports results of computational and experimental studies of the interaction of rhodomyrtone and the bacterial protein FtsZ. The paper is easy to read, the figures are clear, and the experiments are (mostly) well described. The conclusions are somewhat contradictory, as authors acknowledge, but do not reconcile.
A few comments on things that would improve this paper:
1) Authors do not describe the origin of their reagents. Some are not well described (p9 –“polyethyleneglycol” – what MW? Source?). Others are not standard reagents and source should be described (eg 3-MBA).
2) Authors do not mention availability of raw data. I am not sure if MD trajectories count for this or not.
3) Microscopy used to visualize FtsZ (Fig 6) is not described though the microscope is (are these phase contrast images, eg?).
4) The rhodomyrtone binding site is found to be in the nucleotide binding area, predicting GTP competition, but this is not verified by experiment (rhodomyrtone inhibits GTPase activity only partially) . Authors acknowledge this, but make no attempt to explain it (see next point also)
5) A binding affinity of 35+ kcal/mol is reported, but this extremely high affinity is not explored (is this at all reasonable? How does the derived binding affinity for GDP (also very high at 24 kcal/mol) compare to measured values? The value of 35 kcal/mol is very high (avidin-biotin binding is ~ 20 kcal/mol). This requires explanation and discussion by the authors, and on two points: what does this value mean? Even if these values are not comparable to absolute measurements for some reason, are the relative values of GDP and rhodomyrtone comparable? If not, what is the point? If so, how might the incomplete inhibition of GTPase activity be explained? I’m not suggesting that the authors must come up with an absolute explanation of this. But currently the derived numbers are just thrown out there without context (e.g. comparison to other ligands such as avidin-biotin) or explanation.
6) On line 380, it states that Z-ring formation was not as efficiently inhibited by rhodomyrtone. As efficiently as what? Fig 8 shows that dispersion of FtsZ is perhaps the most dramatic effect presented in the paper (at least to that point). Minor point, but the wording is incorrect (“not as efficiently” requires “as __X__”), and I did not understand why authors downplayed this result.
7) Figure 8 legend refers to error bars. I think this was meant to be part of Fig 7.

Reviewer 2 ·

Basic reporting

The authors investigate the specific effects of a natural antimicrobial compound, rhodomyrtone, on FtsZ from gram-positive bacteria. Rhodomyrtone is proven to be as effective as last-resort antibiotics in vivo, including MRSA and vancomycin-intermediate S. aureus strains. The exact target of rhodomyrtone has not been determined, but proteomic and transcriptomic analysis in other studies suggested that the cytoplasmic membrane and cell wall are targeted. In silico screening by this same group for targets identified FtsZ, and therefore the authors investigate the specific effects on FtsZ in silico, in vitro, and in vivo. The authors clearly express their findings with a linear thought process that is easy to follow. The manuscript appears to be in the PeerJ standard format and figures are well presented.

Experimental design

The research question was well-defined, but the path of the researchers seems at times misdirected. The study lacks adequate controls in several experiments. This study is designed because of the results obtained from this group in a previous in silico screen for rhodomyrtone targets.
Comments
1. Fig. 3A- Please insert a referral to Fig. 3A, which is not referenced in the text. In the figure, the labels on the structure are problematic and do not clearly indicate what is being highlighted. Please move the label text off to the side and include arrows. This is especially distracting for the nucleotide binding site label. Fig. 3E and 3F, show the presumed binding site/docking of rhodomytone on the structures. Line 285- did you mean “binding site”? Fig. 3 title has FtsZ misspelled. In the Fig. 3A legend, please add the reference for pdb 3V0A.

2. Fig. 4; Change title to “Structural model of rhodomyrtone-bound FtsZ”

3. Fig. 6; If an average spectra is shown, also include error at each point, otherwise show a replicate that is representative of at least 3 data sets.

4. Line 291 – typo, please change to “evidenced”

5. Line 294 – pleas add “In the simulated model, (S)-rhodomyrtone interacts with…”

6. Line 320 – The statement that “the results indicate that (S)-rhodomyrtone might affect FtsZ assembly dynamics” is over-reaching. The results simply indicate that there could be more flexibility in FtsZ under the conditions.

7. It is unclear whether rhodomytone reduces bundling or prevents polymerization and the light scattering does not differentiate. Also, the authors do not establish a clear concentration-dependent effect by rhodomytone. This should be performed by both light scattering and sedimentation assays comparing supernatants and pellets. Line 339 – as microscopy is not quantitative, the method does confirm that rhodomytone affects the “degree of bundling”. Please remove this language.

8. Fig. 7. The plot of Percent reduction is misleading and it is not clear what the actual untreated rate is in this assay. Replot this using measured rates or amount of pmol or nmol phosphate detected (comparison to an inorganic phosphate standard curve).

9. In the “Morphological changes” section, the authors state that Z-ring formation was not efficiently inhibited by rhodomyrtone in vivo, however, it is, but in a different manner than expected or different than the effect of 3-MBA. The authors later confirm in this section that excess rhodomyrtone does not cause a traditional cell division phenotype of cell elongation, which changes their hypothesis.

10. After the in silico results sections, the authors do not specify if they are using (S)- or (R)-rhodomyrtone for the remainder of the experiments and should clarify where appropriate. I would assume (S) since they observed the change in conformation in silico, but I would have tested both experimentally regardless. Were both tested, or just one type, or a mix?

Validity of the findings

After dismissal of FtsZ as a target in vivo (even though evidence suggests that it is a target), there is a lack of pursuit or discussion of other potential targets or follow up experiments. The discussion should include more explanations for how rhodomyrtone may act on FtsZ in vivo or the cell membrane or both. It is interesting that the difference in conformational changes between Methods are described with sufficient detail and there is speculation where appropriate in the manuscript. The authors could elaborate on the potential mechanism of rhodomyrtone, since there is still data to suggest that it influences FtsZ enzymatic activity in silico and in vitro. Only the last figures support the final conclusions, which means the authors should further pursue the new direction of this research or provide more complete explanations for rhodomyrtone’s effect on gram-positive bacteria.

1. Lines 357-362. The logic in this paragraph is confusing, specifically line 357. If rhodomyrtone competes for the GTP binding site of FtsZ, then FtsZ with GTP would be unable to polymerize in the presence of excess rhodomyrtone. However, polymer bundles are detected in Fig. 6B. It is likely that binding to the GTP binding site would preclude polymerization.

2. Line 366 – FtsZ and FtsA are the first proteins to localize to midcell.

3. Line 371 and Fig. 8. The cells in D, E, and F clearly show that rhodomyrtone is disrupting Z-ring assembly. Presumably the mechanism is completely different from 3-MBA, which stabilizes polymers and prevents their disassembly. It seems presumptive to suggest that the impaired localization is due to FtsA and/or membrane depolarization; however, this could be tested by comparing the effects of several ionophores, including CCCP in this system to see if there is a similar disruption of Gfp-FtsZ. Of note, in Strahl, et al, MinD was also mislocalized. It is more likely that Gfp-FtsZ is unable to assemble properly, bundle or or maintain protein interactions, and these effects are contributing to impaired Z-ring assembly. As Z-ring formation and recruitment of cell wall insertion/remodeling enzymes are linked, it is not too surprising that cells are misshapen (as in Fig. 9). Another possibility is that treatment of the cells results in activation of some cellular proteases, which could result in cleavage of the Gfp- moiety from FtsZ or partial degradation of the fusion – a western blot using Gfp specific antibodies should be performed to confirm that Gfp-FtsZ is still intact.

4. Fig. 9 - Please report cell lengths for all conditions in Fig. 9 by measuring at least 100 cells.

5. Fig. 414-416 – rhodomyrtone treatment leads to impaired Z-ring formation by Gfp-FtsZ. It seems premature to conclude that rhodomyrtone leads to similar effects as CCCP, especially since cells treated with CCCP do not resemble cells treated with rhodomyrtone. B. subtilis cells were reported to have morphological changes upon oxygen depletion, however impaired PBPs or misdirected cell wall insertion and remodeling would be more likely. The actions of pbps and their role in cell shape should be discussed further.

6. Line 443. The conclusion that FtsZ is not the main in vivo target but rather the cell envelope is, is not supported. There is support that FtsZ and potentially other cell wall remodeling enzymes may be the targets of rhodomyrtone action in vivo.

Comments for the author

Additional edits
Please remove the reference to Fig 1 from Material and Methods to the Results section (line 251).
Line 107 …GDP-FtsZ, GTP-FtsZ, (S)-rhodo… (“GTP-FtsZ” was twice in a row)
Line 146 …down into: (no colon)
Line 149 electrostatic (remove “s,” not supposed to be plural)
Line 181 …and 1 complete mini protease inhibitor tablet (“tablet” at end)
Line 204 fluorometer (misspelled)
Line 207 polyethylene glycol (misspelled, supposed to be 2 words)
Line 255 (need to reference Fig. 1 somewhere – not referenced)
Line 270 (need to reference Fig. 3A somewhere – not referenced)
Line 285 (Fig. 4A is referenced before 3F – out of order)
Line 296 residues of FtsZ. (delete “amino acid,” add “FtsZ”)
Line 340 nonspecific (misspelled)
Line 370 (Fig. 8G-H are referenced before 8C-D – change order in the figure or text to present the data in order)
Line 391 (delete “regulated”)
Line 433 (delete “could”)

---

## Round 0.2 · accepted · Accept

· Academic Editor

Accept

I am pleased to accept your manuscript for publication. I agree with reviewer #1's request for theninclusion of the molecular mass of the PEG you used, but I believe that small modification can be addressed directly with PeerJ's production staff

Reviewer 1 ·

Basic reporting

no comment

Experimental design

no ocmment

Validity of the findings

Description of results in this revision is much improved.

Comments for the author

The additional comments and discussion that authors have added resolve the conflicts between the calculations and the results, even if it does still leave unclear the significance of calculated binding affinities (eg drug vs GDP/GTP).
The only thing that I thought should be clarified and wasn't was using PEG as a crowding agent. I asked before that authors clarify what "PEG" meant, and they supplied the name of the supplier from whom they obtained the material. While this is good, the ability of PEG to act as a crowder is very MW-dependent. Hence what I was really asking for is whether the PEG that they used was 2 kDa, 8 kDa, 20 kDa, 50 kDa, etc.

Reviewer 2 ·

Basic reporting

no comment

Experimental design

no comment

Validity of the findings

no comment

Comments for the author

Incorporation of additional experiments, interpretation and analysis provide valuable insight.